# BRAF-AXL-PD-L1 Signaling Axis as a Possible Biological Marker for RAI Treatment in the Thyroid Cancer ATA Intermediate Risk Category

**DOI:** 10.3390/ijms241210024

**Published:** 2023-06-12

**Authors:** Cristina Pizzimenti, Vincenzo Fiorentino, Antonio Ieni, Esther Diana Rossi, Emanuela Germanà, Luca Giovanella, Maria Lentini, Ylenia Alessi, Giovanni Tuccari, Alfredo Campennì, Maurizio Martini, Guido Fadda

**Affiliations:** 1Dipartimento di Scienze Biomediche, Odontoiatriche e Delle Immagini Morfologiche e Funzionali, Divisione di Medicina Nucleare, Università Degli Studi di Messina, 98125 Messina, Italy; cristina.pizzimenti@unime.it (C.P.); emanuelagermana@hotmail.it (E.G.); alfredo.campenni@unime.it (A.C.); 2Dipartimento di Patologia Umana Dell’adulto e Dell’età Evolutiva Gaetano Barresi, Divisione di Anatomia Patologica, Università Degli Studi di Messina, 98125 Messina, Italy; vincenzo.fiorentino@unime.it (V.F.); antonio.ieni@unime.it (A.I.); maria.lentini@unime.it (M.L.); ylenia.alessi91@gmail.com (Y.A.); giovanni.tuccari@unime.it (G.T.); guido.fadda@unime.it (G.F.); 3Dipartimento di Scienze Della Salute e Salute Pubblica, Divisione di Anatomia Patologica, Università Cattolica del Sacro Cuore, Fondazione Policlinico A. Gemelli, IRCCS, 00168 Roma, Italy; esther.rossi@unicatt.it; 4Ente Ospedaliero Cantonale, Istituto Imaging della Svizzera Italiana, Clinica di Medicina Nucleare e Imaging Molecolare, 6500 Bellinzona, Switzerland; luca.giovanella@eoc.ch

**Keywords:** thyroid cancer, BRAF mutation, PD-L1 expression, AXL expression

## Abstract

The use of radioiodine therapy (RIT) is debated in intermediate-risk differentiated thyroid cancer (DTC) patients. The understanding of the molecular mechanisms involved in the pathogenesis of DTC can be useful to refine patient selection for RIT. We analyzed the mutational status of BRAF, RAS, TERT, PIK3 and RET, and the expression of PD-L1 (as a CPS score), the NIS and AXL genes and the tumor-infiltrating lymphocytes (TIL, as the CD4/CD8 ratio), in the tumor tissue in a cohort of forty-six ATA intermediate-risk patients, homogeneously treated with surgery and RIT. We found a significant correlation between BRAF mutations and a less than excellent (LER, according to 2015 ATA classification) response to RIT treatment (*p* = 0.001), higher expression of the AXL gene (*p* = 0.007), lower expression of NIS (*p* = 0.045) and higher expression of PD-L1 (*p* = 0.004). Moreover, the LER patient group had a significantly higher level of AXL (*p* = 0.0003), a lower level of NIS (*p* = 0.0004) and a higher PD-L1 level (*p* = 0.0001) in comparison to patients having an excellent response to RIT. We also found a significant direct correlation between the AXL level and PD-L1 expression (*p* < 0.0001) and a significant inverse correlation between AXL and NIS expression and TILs (*p* = 0.0009 and *p* = 0.028, respectively). These data suggest that BRAF mutations and AXL expression are involved in LER among DTC patients and in the higher expression of PD-L1 and CD8, becoming new possible biomarkers to personalize RIT in the ATA intermediate-risk group, as well as the use of higher radioiodine activity or other possible therapies.

## 1. Introduction

Standard therapeutic strategies for differentiated thyroid cancer (DTC) range from surveillance to a combination of surgery and radioiodine therapy (RIT), depending on the histotype, the aggressiveness of the neoplasm and the clinical and biochemical course of the disease [1]. 

Although there is a uniform consensus on RIT for patients belonging to the American Thyroid Association (ATA) [2] high-risk DTC category, the use of radioiodine is not recommended for the ATA low-risk category and is still controversial in intermediate cases (approximately 70% of all DTC) [3]. In fact, recent retrospective studies demonstrated the limited utility of radioactive iodine (RAI) in reducing cancer mortality or improving prognosis in the intermediate ATA class and, although RAI therapy is universally considered a safe and well-tolerated treatment, some side effects are possible (in the salivary glands, for example), in addition to the related increase in healthcare costs [3]. Moreover, it is known that in cancers such as lung cancer, RAI promotes an increase in PD-L1 expression, thus weakening the activity of immunosurveillance [4]. On the other hand, several scientific reports showed improved overall survival for RAI-treated intermediate-risk DTC patients with specific clinical and histological subtypes (defined as intermediate–high-risk subclass), such as aggressive papillary thyroid cancer variants or evidence of extrathyroidal extension or an increasing volume of nodal disease [5]. Thus, if RIT is useful for this intermediate–high-risk class, more research is needed to understand the therapeutic efficacy of radioiodine in the ATA intermediate category overall. 

Approximately 10–15% of DTC patients do not respond to RIT and develop radioactive iodine resistance (RAI-R) [6]. The molecular mechanism involved in RAI-R has been deeply investigated, including attempts to overcome it and restore tumor radioiodine sensitivity [7]. The incidence of RAI-R DTCs is higher in tumors that harbor the v-raf murine sarcoma viral oncogene homolog B(BRAF) mutations, which determine the hyperfunction of the mitogen-activated protein kinase (MAPK) and phosphoinositide 3-kinase (PI3K) pathways; these control the expression of the sodium iodide symporter (NIS), a protein required for the active concentration of RAI within the thyroid gland [8,9,10]. In addition, recent data seem to highlight the central role of high AXL (Anexelekto) tyrosine kinase receptor expression in RAI refractoriness and NIS dysfunction, especially when combined with the BRAF V600E mutation [11]. 

According to the ATA guidelines, the response to initial treatment can be scored as excellent (ER) and “less than excellent/incomplete response” (LER), including a biochemically indeterminate/incomplete or structurally incomplete response [1]. In the LER group, some molecular alterations involved in RAI-R could already be present, probably due to the altered biological pathways of the tumor. In this way, the stratification of DTC patients according to the tumor’s molecular features could be useful in the early selection of patients who could most benefit from RIT, especially in the ATA intermediate-risk group and patients who require higher-activity RAI or other treatments. 

## 2. Results

### 2.1. Patients’ Characteristics 

The main clinicopathological characteristics of our cohort (46 patients) are reported in Table 1. The mean age at the time of diagnosis was 46.4 years, and 67.4% of patients were female (31 cases). Thirty-two patients were classified as TNM stage 1 (69.6%), while 14 were classified as stage 2 (30.4%). Thirty-six out of 46 patients (78.3%) had lymph node metastatic disease, while 10 out of 46 (21.7%) patients did not. Thirty-two out of 46 patients (69.6%) showed an ER 12 months after RIT, while fourteen out of 46 patients (30.4%) showed an LER: 11 patients (23.9%) with a structurally incomplete response (SIR) and 3 patients (6.5%) with a biochemically indeterminate/incomplete response (BIR). Mutational analysis showed BRAF mutation (only V600E mutation) in twenty-two out of 46 patients (47.8%), while 24 out of 46 patients (52.2%) did not have this. We also found a TERT mutation in one out of 46 patients (2.2%) and an NRAS mutation in two out of 46 patients (4.3%). Our samples did not show an MMR deficiency, and we did not find PIK3 and RET genetic alterations. PD-L1 IHC was performed using the 22C3 pharmDx kit and the CPS was calculated for each sample (Figure 1): 10 out of 46 patients (21.7%) had a higher level of PD-L1 expression (CPS ≥ 1), while 36 (78.3%) had low PD-L1 levels (CPS < 1). Moreover, we found that 23 out of 46 patients (50%) showed higher expression of the AXL gene, whereas the other half of our cohort had a lower level (50%). Twenty-four out of 46 patients (52.2%) had a low level of NIS expression, while 22 (47.8%) had a high level. Finally, 13 patients (28.3%) showed a low CD4/CD8 ratio in the tumor tissue, in comparison to 33 out of 46 (71.7%) that had a high ratio.

### 2.2. Association between BRAF Mutation and Clinical and Biological Parameters

Correlating the BRAF mutation with the main clinical parameters, we found a significant association between the TNM stage (*p* = 0.009, Table 2) and neoplastic lymph node involvement (*p* = 0.011, Table 2). In contrast, BRAF mutation was not related to age or gender.

Notably, in the LER group, twelve out of 14 patients (85.7%) were BRAF-mutated, while two patients (14.3%) were of the BRAF wild type. In the ER group, ten out of 32 patients (31.2%) were BRAF-mutated, while twenty-two (68.8%) were of the BRAF wild type. We found a significant association between BRAF-mutated patients and LER to RIT with respect to those with an excellent response (85.7% vs. 31.2%, *p* = 0.001; Table 2). Among the LER group, all SIR patients were BRAF-mutated, compared to one out of three in the BIR subgroup (*p* = 0.033).

When we correlated the BRAF mutation with AXL, we found a significant correlation between patients with a high AXL level and a BRAF mutation (*p* = 0.007, Table 2). Contrarily, when we analyzed the expression of NIS, we found a significant correlation between a low level of NIS and a BRAF mutation (*p* = 0.045, Table 2). We also found that BRAF-mutated patients had lower NIS expression, by approximately 2.6 times, in comparison to those without mutations (*p* = 0.002). These results were also confirmed by the direct comparison of AXL and NIS mRNA expression in patients with and without BRAF mutations (*p* = 0.012, AXL had higher expression by approximately two times in BRAF-mutated patients in comparison to those without mutations; Student t-test). 

Finally, considering the cut-off of ≥1, we found a significant correlation between higher expression of PD-L1 and a BRAF mutation (*p* = 0.004, Table 2). In contrast, we did not find a significant association between BRAF mutation and TIL, expressed as the CD4/CD8 ratio, in the tumor tissue.

### 2.3. AXL, NIS, PD-L1 and CD4/CD8 Expression (TIL) and Association with the RAI Response 

Analyzing the AXL expression, we found that the relative mRNA level was significantly higher, by approximately 3.5 times, in LER patients in comparison to ER patients (*p* = 0.0003, Figure 2, panel A). In contrast, LER patients had a significantly lower level of NIS in comparison to ER patients (approximately 3.4 times, *p* = 0.0004, Figure 2, panel B). Moreover, we found that LER patients had a significantly higher PD-L1 level and a lower CD4/CD8 ratio with respect to the ER patients (approximately 3 times and *p* = 0.0001 for PD-L1; approximately 1.5 times and *p* = 0.0332 for CD4/CD8 ratio).

In addition, we found a significant and direct correlation between the AXL level and PD-L1 expression (r = 0.867, *p* < 0.0001, Figure 3, panel A), while NIS and TIL showed a significant inverse correlation with AXL (r = −0.473, *p* = 0.0009 for TIL, Figure 3, panel B; r = −0.468, *p* = 0.028 for NIS, Figure 3, panel C). Finally, TIL had a significant inverse correlation with PD-L1 expression (r = −0.410, *p* = 0.0046, Figure 3, panel D).

Our results were confirmed by an analysis performed on the TCGA database, using the UCSC Xena software (https://xena.ucsc.edu/, accessed on 28 March 2023).

After downloading raw data, we found that BRAF-mutated patients had a significantly higher level of AXL and CD274 (PD-L1), and significantly lower expression of NIS, in comparison to those with the BRAF wild type (*p* < 0.0001 for AXL, *p* < 0.0001 for CD274 and *p* < 0.0001 for NIS; patients were divided into low- and high-expression groups using the median expression level as a cut-off; Figure 4, panel A–C). Moreover, the TCGA database analysis showed a significant direct association between AXL expression and CD274 expression (r = 0.306; *p* < 0.0001). In contrast, the TCGA database analysis showed an inverse and close to significant association between AXL and NIS expression (Spearman r = −0.073; *p* = 0.081).

## 3. Discussion

In this work, we have demonstrated that patients with intermediate-risk thyroid carcinoma, treated with thyroid surgical resection and RIT, have a significantly higher risk of displaying an LER to radioiodine treatment when harboring a BRAF mutation (*p* = 0.001). We also found that BRAF mutations were significantly associated with lower levels of NIS mRNA expression in our cohort (*p* = 0.002). These data were confirmed by the TCGA database analysis (based on a larger cohort of 615 patients; *p* < 0.0001) and were in agreement with the literature data reporting that genetic and epigenetic alterations in RTK/BRAF/MAPK/ERK are both inversely correlated with NIS expression and directly correlated with dedifferentiation, recurrence and metastasis [3,7,12]. Interestingly, all patients with SIR, which is associated with significantly worse clinical outcomes than in patients with BIR, showed a BRAF mutation in our cohort [13].

Recently, Collina F. et al. demonstrated a key role of AXL/AKT/NF-kB in the RAI refractoriness and disease persistence or recurrence of thyroid cancer, especially when combined with BRAF mutations [11]. Moreover, they demonstrated that the hyperexpression of AXL significantly reduced the NIS expression and the radioiodine uptake in normal rat thyroid cells. Our analysis also demonstrated that LER patients had a significantly higher level of the AXL gene (*p* = 0.0003) [11]. Moreover, we observed that AXL expression was significantly higher in patients with BRAF mutations (*p* = 0.012) and that NIS expression showed a significant inverse correlation with AXL expression (*p* = 0.028). Our results were also confirmed by an ATLAS database analysis. Our analysis demonstrated that an altered AXL/NIS pathway was only present in the LER group, especially when associated with a BRAF mutation, highlighting that AXL expression levels could be used as predictors of RIT effectiveness. In contrast to Collina F. et al., our data suggest that the upregulation of the AXL gene is probably linked to a BRAF mutation that, through MEK/ERK signaling activation, could promote the AXL mRNA expression, as recently described by others [14].

Although the mutational burden of thyroid cancer is relatively low in comparison to others, defining this tumor as “cold” [15], immunotherapy is considered a novel therapeutic approach, and it has been recently taken into consideration for less differentiated forms of thyroid carcinoma or in those tumors that develop RAI refractoriness, also in association with other drugs such as MAPK inhibitors, PI3K/Akt, other multi-kinase inhibitors (TKI) or other chemotherapy adjuvants, including metformin-modified chitosan (Ch-Met), which could modulate the expression of PD-L1 [15,16,17]. In addition, the high tolerability of immunotherapy compared to chemotherapy and targeted therapies, and recent studies on thyroid cancer microenvironment aimed at identifying tumors that may be more susceptible to immunotherapy [18], have aroused much interest in considering this therapy for DTC. In our analysis, we found that PD-L1 expression had a significantly higher level in LER patients (*p* = 0.0001), a significant association with BRAF mutations (*p* = 0.004) and a direct and significant correlation with higher levels of AXL (*p* < 0.0001). These results were confirmed by the TCGA database analysis and were not related to MSI-MMR deficiency, because no samples from our cohort showed a defect in this pathway. At the same time, we demonstrated a significant correlation between a higher PD-L1 level and lower TIL (CD4/CD8 ratio; *p* = 0.0046). Our results are reinforced by other studies that have reported that BRAF mutations are associated with higher expression of the PD-L1 protein in thyroid cancer, and that increased PD-L1 expression is significantly associated with disease recurrence and poor survival [19,20]. The direct and significant association between BRAF mutations and a higher level of PD-L1 expression in thyroid cancer, as also described in other human cancers, such as NSCLC, is probably due to a relatively higher TMB level [21,22] or the induction of epithelial/mesenchymal (EM) transition with subsequent greater tumor immune evasion, as a consequence of AXL-PI3Kinase-PD-L1 signaling axis activation, a mechanism recently described in head and neck and lung cancer [23,24]. Nonetheless, the presence of higher levels of CD8+ in the tumor microenvironment could be an index of a patient’s better immune response, and, consequently, of the higher PD-L1 expression, indicating the higher intrinsic immune escape capability of the tumor [25].

Finally, our analysis seems to support the idea that some molecular alterations (BRAF mutations and AXL expression, for example) could also have the potential to refine the risk of recurrence re-stratification in DTC treatment (dynamic risk assessment), mainly when interpreted in the context of other clinicopathological risk factors, to provide a more accurate prediction of the status at final follow-up and a more individualized approach [26].

Our study had some limitations. The first was the low number of cases analyzed, and the fact that they were homogeneously classified as ATA intermediate and homogeneously treated with surgery and RAI, notwithstanding the confirmatory analysis of many of our results using the TCGA database. The second was the retrospective design of our study; consequently, further prospective studies are needed to confirm our data and definitively support our findings.

## 4. Materials and Methods

### 4.1. Study Design and Sample Collection

The present study represented an observational retrospective study of 46 patients with ATA intermediate-risk thyroid carcinoma who underwent total thyroidectomy and subsequent RIT at our institution (Azienda Ospedaliera Policlinico Universitario “G. Martino”, Messina). Enrolled patients were diagnosed according to the WHO classification of tumors of the endocrine system [27]. Fixation of the specimens was performed using 10% buffered formalin with exposure ranging from 12 to 48 h. Then, surgically resected samples were paraffin-embedded and, for each case, a representative hematoxylin and eosin (H&E)-stained slide was obtained. All patients underwent total thyroidectomy, and therapeutic neck dissection was performed as clinically indicated (biopsy-proven lymph node metastases, suspicious findings on preoperative neck ultrasound or advanced primary lesions noted during operation). 

RIT was initiated approximately 4 to 6 months after surgery. Before administration of RAI, patients were prepared by levothyroxine withdrawal along with a low-iodine diet for at least 3 weeks to achieve an appropriate TSH level above 30 mIU/L. For patients with intermediate risk, RIT was performed with an adjuvant purpose and using fixed activity of 2220 MBq. Levothyroxine was given on the third day after RAI administration. The response to RIT treatment was evaluated 12 months after RAI administration. 

### 4.2. Follow-Up Strategy and Clinical Outcome

After RIT, patients were regularly followed up by annual thyroglobulin (Tg), thyroglobulin antibody (Tg-Ab) and TSH measurement, a diagnostic whole-body scan and neck ultrasound. If Tg (Tg cut-off of 0.02 ng/dL) was converted from negative to positive or showed sustained growth, or if suspected positive imaging findings were observed, additional imaging methods were implemented. In addition, further RAI therapy was provided.

Based on the comprehensive imaging results and serological results, the ongoing risk stratification of the 2015 ATA guidelines was used to evaluate the clinical outcome and response to RAI therapy at the end of follow-up. The response to initial treatment was scored as excellent (ER) or “less than excellent/incomplete response” (LER), including a biochemically indeterminate/incomplete or structurally incomplete response. All patients’ data were collected anonymously, and written informed consent, as part of the routine diagnosis and treatment procedures, was obtained from patients or their guardians according to the Declaration of Helsinki; the study adhered to the Good Clinical Practice guidelines.

### 4.3. Inclusion Criteria 

We included adults aged 18 years or older, diagnosed with papillary thyroid cancer between January 2019 and December 2022, who underwent total thyroidectomy. Patients were required to display papillary thyroid cancer histology as assessed following the WHO classification of tumors of the endocrine system. All patients were classified into the intermediate-risk DTC category (ATA Management Guidelines, 2009), and all patients were subjected to radioiodine treatment. Patients with aggressive variants of papillary thyroid cancer, such as tall cell, columnar, sclerosing and insular variants or Hürthle cell, and medullary and other poorly differentiated thyroid cancers, or evidence of extrathyroidal extension or an increasing volume of nodal disease were excluded. Patients with more than one primary cancer were also excluded. Additional cases were excluded due to missing clinical and molecular data (such as TNM stage or tumor tissue unavailable, for example).

The study was carried out in accordance with the Declaration of Helsinki, and consent to the retrospective analysis of all clinical data, according to the Ethical Committee of the University of Messina (AOU, “G. Martino” Hospital), was obtained from all the patients (Prot. 65-22) [28]. The report does not present identifying images or other personal or clinical details of participants that compromise their anonymity. 

### 4.4. BRAF, NRAS, TERT, PIK3 and RET Mutational Analysis 

After DNA and RNA extraction from paraffin-embedded tissue, using the QIAcube system (Qiagen, Hilden, Germany) and AllPrep DNA/RNA FFPE Kit (Qiagen), the mutational status of the BRAF, NRAS, PIK3 and RET genes was determined using the Myriapod NGS Cancer Panel DNA, REF NG033, Kit IVD (Diatech Pharmacogenetics, Jesi, Italy) on the EasyPGX qPCR Instrument 96 (Diatech Pharmacogenetics), and the sequencing was performed using iSeq100 (Diatech-Illumina). TERT mutational analysis was performed as previously described [29,30]. 

### 4.5. Real-Time PCR for NIS and AXL Expression

RNA was treated with RQ1 RNase-free deoxyribonuclease (Promega, Milan, Italy) and then the levels of NIS and AXL mRNA were assessed by real-time PCR using SYBR green chemistry. Diluted (1/20) complementary DNA (4 μL) was added to a PCR mix containing 8.4 μL sterile water, 12.5 μL 2 × SYBR mix (Qiagen) and 0.05 μL each of the forward and reverse primers (200 mM) to achieve a final volume of 25 μL. Cycling conditions were 95 °C for 5 min, followed by 40 cycles of 95 °C for 10 s, 60 °C for 30 s and 72 °C for 30 s, and 80 cycles of 55 + 0.5 °C per cycle for melting curve analysis in a CFX96 Real-Time PCR Detection System (Bio-Rad, Milan, Italy). Each assay was performed in triplicate, and data were processed by the CFX Manager software (Bio-Rad). The average obtained for NIS and AXL was normalized to the average amount of β-actin for each sample to determine relative changes in mRNA expression. Using the median relative expression as a cut-off, we identified patients with high or low gene expression when the relative gene level was above or below the cut-off, respectively. Primers used for NIS, AXL and β-actin were as follows: AXL sense 5′-AAC CTT CAA CTC CTG CCT TCT CG-3′ and antisense 5′-CAG CTT CTC CTT CAG CTC TTC AC-3′; NIS sense 5′-CCA TCC TGG ATG ACA ACT TGG-3′ and antisense 5′-AAA AAC AGA CGA TCC TCA TTG GT-3′; β-actin sense 5′-AGC ACT GTG TTG GCG TAC AG-3′ and antisense 5′-AGA GCT ACG AGC TGC CTG AC-3′ [31]. Using the median relative expression as a cut-off, we identified patients with high or low gene expression when the relative gene level was above or below the cut-off, respectively.

### 4.6. PD-L1 Immunohistochemistry (IHC) and Interpretation; Tumor-Infiltrating Lymphocytes (TIL) as CD4 and CD8 Ratio

The PD-L1 immunohistochemistry assay was performed on each specimen (3-μm-thick consecutive sections) with an anti–PD-L1 antibody (clone 22C3), according to the manufacturer’s instructions. Briefly, we used 22C3 pharmDx (mouse monoclonal primary anti-PD-L1 antibody, prediluted, clone 22C3, Dako, Carpinteria, CA, USA) on the Autostainer Link 48 with the EnVision DAB Detection System (Agilent Technologies, Santa Clara, CA, USA). PD-L1 control slides from 22C3 pharmDx (containing sections of two pelleted, formalin-fixed, paraffin-embedded cell lines: NCI-H226 with moderate PD-L1 protein expression and MCF-7 with negative PD-L1 protein expression) were used as positive and negative controls. We also used placenta, tonsil and vermiform appendix tissues as positive controls. The combined positive score (CPS) was determined as the number of PD-L1-positive tumor cells, lymphocytes and macrophages divided by the total number of viable tumor cells, multiplied by 100. Any perceptible and convincing partial or complete linear membrane staining of viable tumor cells that was perceived as distinct from cytoplasmic staining was considered as positive PD-L1 staining and included in the scoring. Likewise, any membrane and/or cytoplasmic staining of mononuclear inflammatory cells within tumor nests and/or adjacent supporting stroma was considered positive PD-L1 staining and was included in the CPS numerator. Neutrophils, eosinophils, plasma cells and immune cells (ICs) associated with in situ components, benign structures or ulcers were excluded from the CPS score. The cut-off of ≥1 was considered. Each countable section contained at least 100 viable neoplastic cells [25]. TILs were evaluated as the CD4+/CD8+ T-cell ratio. The expression of CD4+ and CD8+ was assessed by immunohistochemistry using the anti-CD4 monoclonal antibody (Clone 4B12; Dako, Carpinteria, CA, USA) and anti-CD8 monoclonal antibody (Clone 1A5; Ventana Inc. Tucson, AZ, USA), as previously described, with a few modifications [25].

### 4.7. Immunohistochemistry and Immunostaining Analysis for Mismatch Repair System (MMR)

After fixation in buffered formalin (>48 h) and paraffin embedding, tumor tissue sections were hematoxylin–eosin (HE) stained. The number of tumor nuclei and the contents of cancer cells were evaluated by a pathologist (M.M.). All immunohistochemical analyses were performed using the automated Bond Max immunostainer (Leica Microsystems, Ballerup, Denmark). Whole 3-μm tissue sections were dewaxed and rehydrated. Antigen retrieval was performed using Bond™ Epitope Retrieval Solution 2 (AR9640, Leica Microsystems) at 99 °C for 20 min. After endogenous peroxide blocking, tissue sections were incubated for 30 min with antibodies against MLH1 (rabbit monoclonal anti-MLH1, ab214441, Abcam, Milan, Italy), MSH2 (rabbit monoclonal anti-MSH2, ab212188, Abcam), MSH6 (rabbit monoclonal anti-MLH6, ab208940, Abcam) and PMS2 (rabbit monoclonal anti-PMS2, ab110638, Abcam). Subsequently, primary antibody binding to the sections was detected using the Bond™ Polymer Refine Detection Kit (DS9800, Leica Microsystems). The slides were incubated for 15 min at room temperature with a secondary antibody (mouse anti-rabbit IgG) and tertiary reagent (anti-mouse Poly-HRP-IgG). The slides were incubated in DAB solution for 10 min at room temperature. Each slide was counterstained for 1 min with hematoxylin. MMR protein expression was defined as negative if staining was absent or present in less than 10% of tumor nuclei in a section. Normal epithelial cells and stromal cells were used as internal controls. Thyroid cancer samples were classified as MMR-deficient if one or more of the four proteins showed negative staining.

### 4.8. TCGA Analysis of AXL, NIS and CD274 Expression 

The expression of AXL, NIS and CD274 in thyroid cancer was analyzed using the database of The Cancer Genome Atlas (TCGA), using the University of California Santa Cruz’s Xena software (https://xena.ucsc.edu/, accessed on 28 March 2023). 

### 4.9. Statistical Analysis

Statistical analysis was performed using the MedCalc or StatView ver 5.0 software. Statistical comparison of continuous variables was performed by the Mann–Whitney U-test (*t* test), as appropriate. Comparison of categorical variables was performed with the chi-square statistic, using Fisher’s exact test. Spearman’s rho correlation coefficient (r) was employed to evaluate the associations between two or more variables. *p*-values less than 0.05 were considered statistically significant.

## 5. Conclusions

Our analysis demonstrated that several altered pathways are involved in the LER to RIT, likely due to the tumor’s intrinsic molecular alterations. Moreover, our data suggest that the assessment of BRAF mutations and AXL expression could have twofold value: (1) as a useful biological parameter in the dynamic risk assessment; (2) to identify patients who could be treated with higher-activity RAI or with other possible therapies, such as immunotherapy, mainly in those with higher expression of PD-L1. 

## Figures and Tables

**Figure 1 ijms-24-10024-f001:**
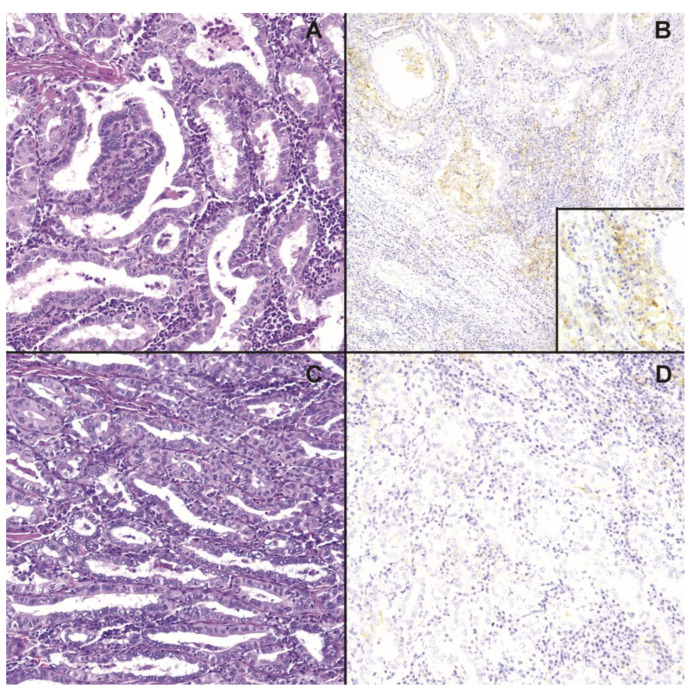
The figure shows two example thyroid cancer cases (panels (**A**) and (**C**), respectively, E&E, 100× magnification) analyzed with 22C3 pharmaDx kit (panels (**B**) and (**D**), respectively; 100× magnification), having higher PD-L1 expression (CPS > 1) in panel (**B**) and lower PD-L1 expression (CPS < 1) in panel (**D**). The superimposed image in panel (**B**) represents a 200× magnification to enable the better visualization of the PD-L1 membrane positivity in tumor cells and tumor-related lymph monocytes.

**Figure 2 ijms-24-10024-f002:**
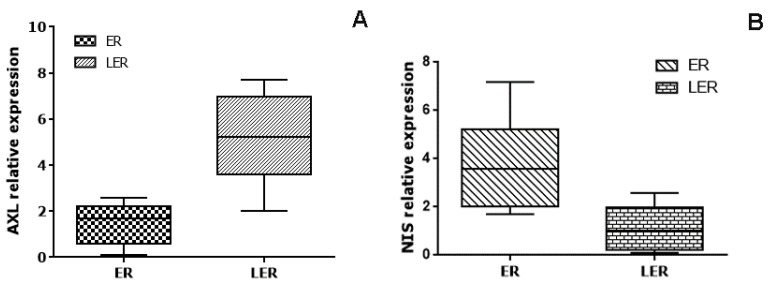
Panel (**A**) and panel (**B**) show the whisker plots of the relative expression of AXL and NIS, respectively, in patients with an excellent (ER) and less than excellent response (LER) to RIT therapy. LER patients had significantly higher expression of AXL (*p* = 0.0003). In contrast, LER patients had a significantly lower level of NIS (*p* = 0.0004).

**Figure 3 ijms-24-10024-f003:**
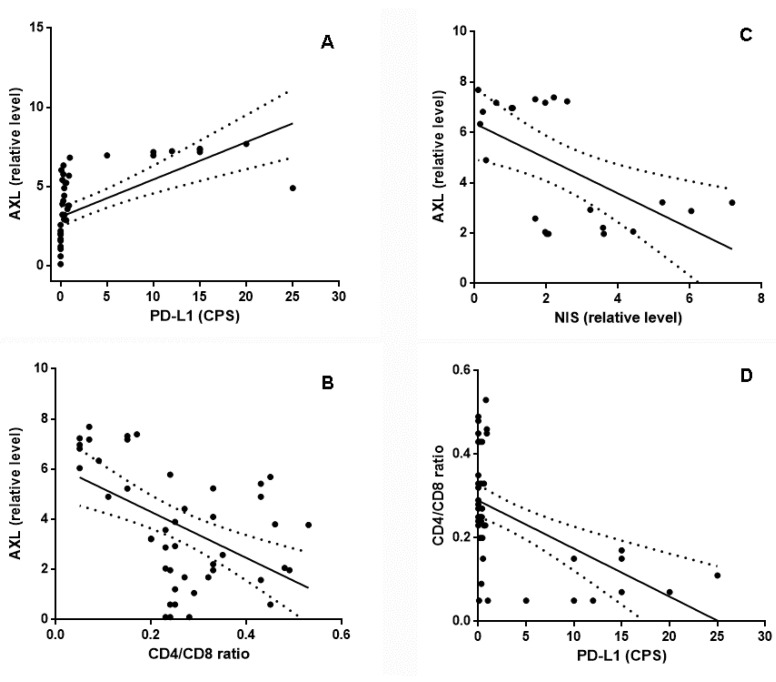
Panel (**A**) shows that the relative AXL level had a significant direct correlation with PD-L1 expression, as the CPS (Spearman r = 0.867; *p* < 0.0001, panel (**A**)), and a significant inverse correlation with TIL (as the CD4/CD8 ratio, Spearman r = −0.473; *p* = 0.0009, panel (**B**)) and NIS expression (Spearman r = −0.468; *p* = 0.028, panel (**C**)). Panel (**D**) shows the significant inverse correlation between PD-L1 expression (as the CPS) and TIL (as the CD4/CD8 ratio, Spearman r = −0.410; *p* = 0.0046).

**Figure 4 ijms-24-10024-f004:**
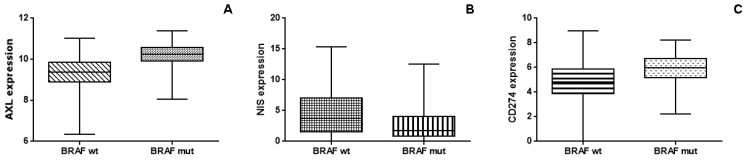
Panels (**A**–**C**) show the whisker plots of the relative expression of AXL, NIS and CD274, respectively, in BRAF-mutated and BRAF-wild-type thyroid cancer from TCGA database. BRAF-mutated thyroid cancer has a significantly higher level of AXL (panel (**A**); *p* < 0.0001) and PD-L1 (panel (**C**); *p* < 0.0001) and significantly lower expression of NIS (panel (**B**); *p* < 0.0001).

**Table 1 ijms-24-10024-t001:** Patient characteristics.

	*n* = 46
Age, mean (±SD)	46.4 (9.7)
Gender, *n* (%)	
Male	15 (32.6)
Female	31 (67.4)
TNM stage, *n* (%)	
I	32 (69.6)
II	14 (30.4)
Lymph node metastasis, *n* (%)	
Positive	36 (78.3)
Negative	10 (21.7)
PD-L1 expression, *n* (%)	
Positive (CPS ≥ 1)	10 (21.7)
Negative (CPS < 1)	36 (78.3)
RAI response (12 months), *n* (%)	
ER	32 (69.6)
LER	14 (30.4)
BRAF mutation, *n* (%)	
Mutated	22 (47.8)
Wild type	24 (52.2)
AXL expression, *n* (%)	
High	23 (50)
Low	23 (50)
NIS expression, *n* (%)	
High	22 (47.8)
Low	24 (52.2)
CD4/CD8 ratio, *n* (%)	
High	33 (71.7)
Low	13 (28.3)

**Table 2 ijms-24-10024-t002:** Relation between BRAF mutational status and clinicopathological features.

	BRAF Mutated	BRAF Wild Type	*p*	OR (95% CI)
**Age**				
**<45**	12	6	0.069	3.600
**≥45**	10	18		From 1.033 to 12.55
**Gender**				
**Male**	9	6	0.348	2.077
**Female**	13	18		From 0.592 to 7.291
**TNM stage**				
**I**	11	21	** 0.009 **	0.143
**II**	11	3		From 0.033 to 0.622
**Lymph node metastasis**				
**Positive**	21	15	** 0.011 **	12.6
**Negative**	1	9		From 1.438 to 110.4
**RAI response**				
**ER**	10	22	** 0.001 **	13.20
**LER**	12	2		From 2.476 to 70.37
**AXL expression**				
**High**	16	7	** 0.007 **	6.476
**Low**	6	17		From 1.788 to 23.45
**PD-L1 expression**				
**Positive (CPS ≥ 1)**	9	1	** 0.004 **	0.063
**Negative (CPS < 1)**	13	23		From 0.007 to 0.553
**NIS expression**				
**High**	7	15	** 0.045 **	0.280
**Low**	15	9		From 0.083 to 0.948
**CD4/CD8 ratio**				
**High**	13	20	0.103	3.462
**Low**	9	4		From 0880 to 13.62

## Data Availability

The datasets used and/or analyzed during the current study are available from the corresponding author on reasonable request.

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
