# Peer review of "BRAF-AXL-PD-L1 Signaling Axis as a Possible Biological Marker for RAI Treatment in the Thyroid Cancer ATA Intermediate Risk Category"

_ijms, 2023, doi:10.3390/ijms241210024_

Round 1

Reviewer 1 Report

The revised manuscript is well written, clearly shows the aim, methods, results and discussion. The problem with different responses to RIT is important and neovel, well described parameters will help with the anti-cancer treatment.

Presentation of the results- I would suggest to change the panel C,D anf F on Fig. 2, in this form are not clear.

In spite of low number of cases the data might be useful and encouraging for further research.

Author Response

Reviewer’s comments, 1

The revised manuscript is well written, clearly shows the aim, methods, results and discussion. The problem with different responses to RIT is important and neovel, well described parameters will help with the anti-cancer treatment.

Thank you for the enthusiastic evaluation of the manuscript.

Presentation of the results- I would suggest to change the panel C,D anf F on Fig. 2, in this form are not clear.

Following the reviewer’s suggestion, we separated the Figure 2 in two new Figures: the Figure 2 showing only the whiskers box of the relative expression of AXL and NIS, respectively, in patients with excellent (ER) and less than excellent response (LER); the a new Figure 3 showing the direct correlation between PD-L1 expression and AXL relative level, and the inverse correlation between AXL relative level and TIL (as CD4/CD8 ratio), between AXL relative level and NIS expression and between PD-L1 expression, as CPS, and TIL (as CD4/CD8 ratio). Moreover, we changed the format of the scatter diagram in Figure 3 for all panels, to make the analysis clearer.

In spite of low number of cases the data might be useful and encouraging for further research.

Reviewer 2 Report

      In this research, the authors researched the possibility of using BRAF-AXL-PD-L1 signaling axis as a possible biological marker for RAI treatment in the thyroid cancer ATA intermediate risk category. Generally, it’s meaningful and interesting research. In my opinion, the current version of this manuscript fits the scope of International Journal of Molecular Sciences and could be accepted after major revision.

My specific comments are in detail listed below:

1.     Some minor mistakes exist in this paper, the authors should carefully check it. For example, p<0.0001 should be p < 0.0001; p=0.0001 should be p = 0.0001.

2.     In the introduction part, how surgery and RIT affect PD-L1 expression should be revealed. Some references could be added to this part including 10.1002/advs.202207608.

3.     The quality of almost all the figures still need to be improved. If possible, some clear version should be added.

4.     In the discussion or some other part, if possible, the current development of PD-L1 regulation strategy could be added. Some references could be added to this part including 10.1016/j.ijbiomac.2022.10.167.

5.     The abbreviations should be listed or pointed out at the first place it existed, including BRAF, RAS, TERT, PIK3, RET, PD-L1, NIS, LER, and AXL.

Author Response

Reviewer’s comments, 2

In this research, the authors researched the possibility of using BRAF-AXL-PD-L1 signaling axis as a possible biological marker for RAI treatment in the thyroid cancer ATA intermediate risk category. Generally, it’s meaningful and interesting research. In my opinion, the current version of this manuscript fits the scope of International Journal of Molecular Sciences and could be accepted after major revision.

My specific comments are in detail listed below:

  1. Some minor mistakes exist in this paper, the authors should carefully check it. For example, p<0.0001 should be p < 0.0001; p=0.0001 should be p = 0.0001.

In the new version of the manuscript, we correct the minor mistakes. The revised manuscript was carefully checked for the English correction with the help of native English language speaking person.

  1. In the introduction part, how surgery and RIT affect PD-L1 expression should be revealed. Some references could be added to this part including 10.1002/advs.202207608.

Following the reviewer’s suggestion, in the new version of the manuscript we reported how RIT affect PD-L1 expression (Introduction part). The bibliographic reference has been adequately added.

  1. The quality of almost all the figures still need to be improved. If possible, some clear version should be added.

Following the reviewer's suggestion, we have increased the quality of the figures and we separated the Figure 2 in two new Figures: the Figure 2 showing only the whiskers box of the relative expression of AXL and NIS, respectively, in patients with excellent (ER) and less than excellent response (LER), and a new Figure 3 showing the direct correlation between PD-L1 expression and AXL relative level, and the inverse correlation between AXL relative level and TIL (as CD4/CD8 ratio), between AXL relative level and NIS expression and between PD-L1 expression, as CPS, and TIL (as CD4/CD8 ratio). Moreover, we changed the format of the scatter diagram in Figure 3, to make the analysis clearer.

  1. In the discussion or some other part, if possible, the current development of PD-L1 regulation strategy could be added. Some references could be added to this part including 10.1016/j.ijbiomac.2022.10.167.

Following the reviewer’s suggestion, in the new version of the manuscript we also reported the current development of PD-L1 regulation strategy, using, for example adjuvant therapy (Discussion part). The bibliographic reference has been adequately added.

  1. The abbreviations should be listed or pointed out at the first place it existed, including BRAF, RAS, TERT, PIK3, RET, PD-L1, NIS, LER, and AXL.

As the reviewer suggested, we added an abbreviations list after the Conclusion part, in the new version of the manuscript.

Round 2

Reviewer 2 Report

The current version of this manuscript could be accepted.